# MULTISCALE INVERTIBLE GENERATIVE NETWORKS FOR HIGH-DIMENSIONAL BAYESIAN INFERENCE

## ABSTRACT

High-dimensional Bayesian inference problems cast a long-standing challenge in generating samples, especially when the posterior has multiple modes. For a wide class of Bayesian inference problems equipped with the *multiscale structure* that low-dimensional (coarse-scale) surrogate can approximate the original high-dimensional (fine-scale) problem well, we propose to train a Multiscale Invertible Generative Network (MsIGN) for sample generation. A novel prior conditioning layer is designed to bridge networks at different resolutions, enabling coarse-to-fine multi-stage training. Jeffreys divergence is adopted as the training objective to avoid mode dropping. On two high-dimensional Bayesian inverse problems, MsIGN approximates the posterior accurately and clearly captures multiple modes, showing superior performance compared with previous deep generative network approaches. On the natural image synthesis task, MsIGN achieves the superior performance in bits-per-dimension compared with our baseline models and yields great interpret-ability of its neurons in intermediate layers.

## 1 INTRODUCTION

Bayesian inference provides a powerful framework to blend prior knowledge, data generation process and (possibly small) data for statistical inference. With some prior knowledge $\rho$ (distribution) for the quantity of interest $\boldsymbol{x} \in \mathbb{R}^d$, and some (noisy) measurement $\boldsymbol{y} \in \mathbb{R}^{d_y}$, it casts on $\boldsymbol{x}$ a posterior

$$q(\mathbf{x}|\boldsymbol{y}) \propto \rho(\mathbf{x})L(\boldsymbol{y}|\mathbf{x}), \quad \text{where} \quad L(\boldsymbol{y}|\mathbf{x}) = \mathcal{N}(\boldsymbol{y} - \mathcal{F}(\mathbf{x}); \mathbf{0}, \boldsymbol{\Gamma_\varepsilon}). \tag{1}$$

where $L(\boldsymbol{y}|\mathbf{x})$ is the likelihood that compares the data $\boldsymbol{y}$ with system prediction $\mathcal{F}(\mathbf{x})$ from the candidate $\mathbf{x}$, here $\mathcal{F}$ denotes the forward process. We can use different distributions to model the mismatch $\boldsymbol{\varepsilon} = \boldsymbol{y} - \mathcal{F}(\mathbf{x})$, and for illustration simplicity, we assume Gaussian in Equation 1. For example, Bayesian deep learning generates model predicted logits $\mathcal{F}(\mathbf{x})$ from model parameters $\mathbf{x}$, and compares it with discrete labels $\boldsymbol{y}$ through binomial or multinomial distribution.

Sampling or inferring from $q$ is a long-standing challenge, especially for high-dimensional (high-$d$) cases. An arbitrary high-$d$ posterior can have its importance regions (also called "modes") anywhere in the high-$d$ space, and finding these modes requires computational cost that grows exponentially with the dimension $d$. This intrinsic difficulty is the consequence of "the curse of dimensionality", which all existing Bayesian inference methods suffer from, e.g., MCMC-based methods (Neal et al., 2011; Welling & Teh, 2011; Cui et al., 2016), SVGD-type methods (Liu & Wang, 2016; Chen et al., 2018; 2019a), and generative modeling (Morzfeld et al., 2012; Parno et al., 2016; Hou et al., 2019).

In this paper, we focus on Bayesian inference problems with multiscale structure and exploit this structure to sample from a high-$d$ posterior. While the original problem has a high spatial resolution (fine-scale), its low resolution (coarse-scale) analogy is computationally attractive because it lies in a low-dimension (low-$d$) space. A problem has the multiscale structure if such coarse-scale low-$d$ surrogate exists and gives good approximation to the fine-scale high-$d$ problem, see Section 2.1. Such multiscale property is very common in high-$d$ Bayesian inference problems. For example, inferring 3-D permeability field of subsurface at the scale of meters is a reasonable approximation of itself at the scale of centimeters, while the problem dimension is $10^6$-times fewer.

We propose a Multiscale Invertible Generative Network (MsIGN) to sample from high-$d$ Bayesian inference problems with multiscale structure. MsIGN is a flow-based generative network that can both generate samples and give density evaluation. It consists of multiple scales that recursively

lifts up samples to a finer-scale (higher-resolution), except that the coarsest scale directly samples from a low-$d$ (low resolution) distribution. At each scale, a fixed prior conditioning layer combines coarse-scale samples with some random noise according to the prior to enhance the resolution, and then an invertible flow modifies the samples for better accuracy, see Figure 1. The architecture of MsIGN makes it fully invertible between the final sample and random noise at all scales.

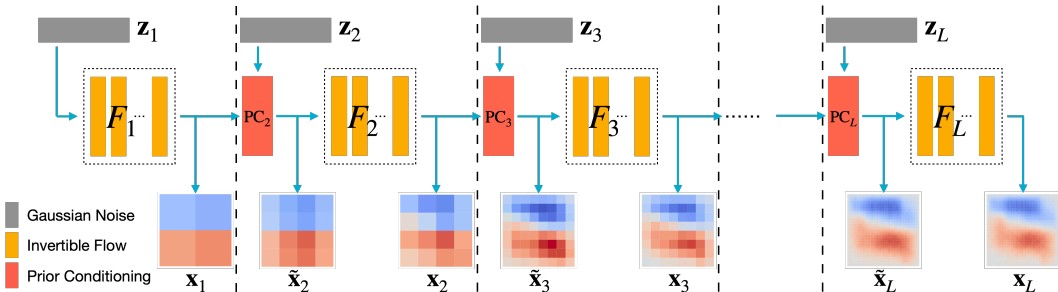

Figure 1: MsIGN generates samples from coarse to fine scale. Each scale, as separated by vertical dash lines, takes in feature $\mathbf{x}_{l-1}$ from the coarser scale and Gaussian noise $\mathbf{z}_l$, and outputs a sample $\mathbf{x}_l$ of finer scale. The prior conditioning layer $\text{PC}_l$ lifts up the coarser-scale sample $\mathbf{x}_{l-1}$ to a finer scale $\tilde{\mathbf{x}}_l$, which is the best guess of $\mathbf{x}_l$ given its coarse-scale value $\mathbf{x}_{l-1}$ and the prior. An invertible flow $F_l$ further modifies $\tilde{\mathbf{x}}_l$ to better approximate $\mathbf{x}_l$. See Section 2.1 for detailed explanation.

MsIGN undergoes a multi-stage training that learns a hierarchy of distributions with dimensions growing from the lowest to the highest (the target posterior). Each stage gives a good initialization to the next stage thanks to the multiscale property. To capture multiple modes, we choose Jeffreys divergence $D_{\text{J}}(p\|q)$ as the training objective at each stage, which is defined as

$$D_{\text{J}}(p\|q) = D_{\text{KL}}(p\|q) + D_{\text{KL}}(q\|p) = \mathbb{E}_{\mathbf{x}\sim p}\left[\log\left(p(\mathbf{x})/q(\mathbf{x})\right)\right] + \mathbb{E}_{\mathbf{x}\sim q}\left[\log\left(q(\mathbf{x})/p(\mathbf{x})\right)\right] . \quad (2)$$

Jeffreys divergence removes bad local minima of single-sided Kullback-Leibler (KL) divergence to avoid mode missing. We build an unbiased estimation of it by leveraging prior conditioning layer in importance sampling. Proper loss function and good initialization from multi-stage training solve the non-convex optimization stably and capture multi-modes of the high-$d$ distribution.

In summary, we claim four contributions in this work. First, we propose a Multiscale Invertible deep Generative Network (MsIGN) with a novel prior conditioning layer, which can be trained in a coarse-to-fine scale manner. Second, Jeffreys divergence is used as the objective function to avoid mode collapse, and is estimated by importance sampling based on the prior conditioning layer. Third, when applied to two Bayesian inverse problems, MsIGN clearly captures multiple modes in the high-$d$ posterior and approximates the posterior accurately, demonstrating its superior performance compared with previous methods via the generative modeling approach. Fourth, we also apply MsIGN to image synthesis tasks, where it achieves superior performance in bits-per-dimension among our baseline models, like Glow (Kingma & Dhariwal, 2018), FFJORD (Grathwohl et al., 2018), Flow++ (Ho et al., 2019), i-ResNet (Behrmann et al., 2019), and Residual Flow (Chen et al., 2019b). MsIGN also yields great interpret-ability of its neurons in intermediate layers.

## 2 METHODOLOGY

We will abbreviate $q(\mathbf{x}|\boldsymbol{y})$ in Equation 1 as $q(\mathbf{x})$ for simplicity in the following context, because $\boldsymbol{y}$ *only* plays the role of defining the target distribution $q(\mathbf{x})$ in MsIGN. In Section 2.1, we discuss the multiscale structure in detail of the posterior $q(\mathbf{x})$ and derive a scale decoupling that can be utilized to divide and conquer the high-$d$ challenge of Bayesian inference.

As a flow-based generative model like in Dinh et al. (2016), MsIGN models a bijective that maps Gaussian noise $\mathbf{z}$ to a sample $\mathbf{x}$ whose distribution is denoted as $p_\theta(\mathbf{x})$, where $\theta$ is the network parameters. MsIGN allows fast generation of samples $\mathbf{x}$ and density evaluation $p_\theta(\mathbf{x})$, so we train our working distribution $p_\theta(\mathbf{x})$ to approximate the target distribution $q(\mathbf{x})$. We present the architecture of MsIGN in Section 2.2 and the training algorithm in Section 2.3.

## 2.1 Multiscale Structure and Scale Decoupling

We say a Bayesian inference problem has *multiscale structure* if the associated coarse-scale likelihood $L_c$ approximates the original likelihood $L$ well:

$$L(\boldsymbol{y}|\mathbf{x}) \approx L_c(\boldsymbol{y}|\mathbf{x}_c)\,, \quad \text{where} \quad L_c(\boldsymbol{y}|\mathbf{x}_c) := \mathcal{N}(\boldsymbol{y} - \mathcal{F}_c(\mathbf{x}_c); \mathbf{0}, \boldsymbol{\Gamma}_{\boldsymbol{\varepsilon}})\,. \tag{3}$$

Here $\mathbf{x}_c \in \mathbb{R}^{d_c}$ is a coarse-scale version of the fine-scale quantity $\mathbf{x} \in \mathbb{R}^d$ ($d_c < d$), given by a deterministic pooling operator $\mathcal{A} : \mathbf{x}_c = \mathcal{A}(\mathbf{x})$. The map $\mathcal{F}_c : \mathbb{R}^{d_c} \to \mathbb{R}^{d_y}$ is a forward process that gives system prediction based on the coarse-scale information $\mathbf{x}_c$. A popular case of the multiscale structure is when $\mathcal{A}$ is the average pooling operator, and $\mathcal{F}(\mathbf{x}) \approx \mathcal{F}_c(\mathbf{x}_c)$, meaning that the system prediction mainly depends on the lower-resolution information $\mathbf{x}_c$. Equation 3 motivates us to define a surrogate distribution $\tilde{q}(\mathbf{x}) \propto \rho(\mathbf{x})L_c(\boldsymbol{y}|\mathcal{A}(\mathbf{x}))$ that approximates the target posterior $q(\mathbf{x})$ well[1]:

$$\tilde{q}(\mathbf{x}) = \rho(\mathbf{x})L_c(\boldsymbol{y}|\mathcal{A}(\mathbf{x})) = \rho(\mathbf{x})L_c(\boldsymbol{y}|\mathbf{x}_c) \approx \rho(\mathbf{x})L(\boldsymbol{y}|\mathbf{x}) = q(\mathbf{x})\,. \tag{4}$$

We also notice that the prior $\rho$ allows an exact scale decoupling. To generate a sample $\mathbf{x}$ from $\rho$, one can first sample its coarse-scale version $\mathbf{x}_c = \mathcal{A}(\mathbf{x})$, and then replenish missing fine-scale details without changing the coarse-scale structure by sampling from the conditional distribution $\rho(\mathbf{x}|\mathbf{x}_c) = \rho(\mathbf{x}|\mathcal{A}(\mathbf{x}) = \mathbf{x}_c)$. Using $\rho_c$ to denote the distribution of $\mathbf{x}_c = \mathcal{A}(\mathbf{x})$, the conditional probability calculation summarizes this scale decoupling process as $\rho(\boldsymbol{x}) = \rho(\boldsymbol{x}|\boldsymbol{x}_c)\rho_c(\boldsymbol{x}_c)$.

Combining the scale effect in the likelihood and the scale decoupling in the prior, we decouple the surrogate $\tilde{q}(\mathbf{x}) = \rho(\mathbf{x})L_c(\boldsymbol{y}|\mathcal{A}(\mathbf{x}))$ into the prior conditional distribution $\rho(\mathbf{x}|\mathbf{x}_c)$ and a coarse-scale posterior, defined as $q_c(\mathbf{x}_c) := \rho_c(\mathbf{x}_c)L(\boldsymbol{y}|\mathbf{x}_c)$. The decoupling goes as

$$\tilde{q}(\mathbf{x}) = \rho(\mathbf{x})L_c(\boldsymbol{y}|\mathbf{x}_c) = \rho(\mathbf{x}|\mathbf{x}_c)\rho_c(\mathbf{x}_c)L_c(\boldsymbol{y}|\mathbf{x}_c) = \rho(\mathbf{x}|\mathbf{x}_c)q_c(\mathbf{x}_c)\,, \tag{5}$$

The prior conditional distribution $\rho(\mathbf{x}|\mathbf{x}_c)$ bridges the coarse-scale posterior $q_c(\mathbf{x}_c)$ and the surrogate $\tilde{q}(\mathbf{x})$, which in turn approximates the original fine-scale posterior $q(\mathbf{x})$. Parno et al. (2016) proposed a similar scale decoupling relation, and we leave the discussion and comparison to Appendix A.

Figure 1 shows the integrated sampling strategy. To sample an $\mathbf{x}$ from $q$, we start with an $\mathbf{x}_c$ from $q_c$. The prior conditioning layer then performs random upsampling from the prior conditional distribution $\rho(\cdot|\mathbf{x}_c)$, and the output will be a sample $\tilde{\mathbf{x}}$ of the surrogate $\tilde{q}$. Due to the approximation $\tilde{q} \approx q$ from Equation 4, we stack multiple invertible blocks for the invertible flow $F$ to modify the sample $\tilde{\mathbf{x}} \sim \tilde{q}$ to a sample $\mathbf{x} \sim q$: $\mathbf{x} = F(\tilde{\mathbf{x}})$. $F$ is initialized as an identity map in training. Finally, to obtain the $\mathbf{x}_c$ from $q_c$, we apply the above procedure recursively until the dimension of the coarsest scale is small enough so that $q_c$ can be easily sampled by a standard method.

## 2.2 Multiscale Invertible Generative Network: Architecture

Our proposed MsIGN has multiple levels to recursively apply the above strategy. We denote $L$ the number of levels, $\mathbf{x}_l \in \mathbb{R}^{d_l}$ the sample at level $l$, and $\mathcal{A}_l : \mathbb{R}^{d_l} \to \mathbb{R}^{d_{l-1}}$ the pooling operator from level $l$ to $l-1$: $\mathbf{x}_{l-1} = \mathcal{A}_l(\mathbf{x}_l)$. Following the idea in Section 2.1, we can define the $l$-th level target $q_l(\mathbf{x}_l)$ and surrogate $\tilde{q}_l(\tilde{\mathbf{x}}_l)$, and the last-level target $q_L$ is our original target $q$ in Equation 1. The $l$-th level of MsIGN uses a prior conditioning layer $\text{PC}_l$ and an inverse transform $F_l$ to capture $q_l$.

**Prior conditioning layer.** The prior conditioning layer $\text{PC}_l$ at level $l$ lifts a coarse-scale sample $\mathbf{x}_{l-1} \in \mathbb{R}^{d_{l-1}}$ up to a *random* fine-scale one $\mathbf{x}_l \in \mathbb{R}^{d_l}$ following the conditional distribution $\rho(\mathbf{x}_l|\mathbf{x}_{l-1})$. The difference in dimension is compensated by a Gaussian noise $\mathbf{z}_l \in \mathbb{R}^{d_l - d_{l-1}}$, which is the source of randomness: $\mathbf{x}_l = \text{PC}_l(\mathbf{x}_{l-1}, \mathbf{z}_l)$. $\text{PC}_l$ depends only on the prior conditional distribution $\rho(\mathbf{x}_l|\mathbf{x}_{l-1})$, and thus can be pre-computed *independently for different levels* regardless of the likelihood $L$. When the prior is Gaussian and the pooling operators are linear (e.g., average pooling), the prior conditional distribution is still Gaussian with moments specified as follows.

**Lemma 2.1** *Suppose that* $\rho(\mathbf{x}_l) = \mathcal{N}(\mathbf{x}_l; \mathbf{0}, \boldsymbol{\Sigma}_l)$, *and* $\mathcal{A}_l(\mathbf{x}_l) = \boldsymbol{A}_l\mathbf{x}_l$ *for some* $\boldsymbol{A}_l \in \mathbb{R}^{d_{l-1} \times d_l}$, *then with* $\boldsymbol{U}_{l-1} := \boldsymbol{\Sigma}_l\boldsymbol{A}_l^T(\boldsymbol{A}_l\boldsymbol{\Sigma}_l\boldsymbol{A}_l^T)^{-1}$ *and* $\boldsymbol{\Sigma}_{l|l-1} := \boldsymbol{\Sigma}_l - \boldsymbol{\Sigma}_l\boldsymbol{A}_l^T(\boldsymbol{A}_l\boldsymbol{\Sigma}_l\boldsymbol{A}_l^T)^{-1}\boldsymbol{A}_l\boldsymbol{\Sigma}_l$, *we have*

$$\rho(\mathbf{x}_l|\mathbf{x}_{l-1} = \boldsymbol{A}_l\mathbf{x}_l) = \mathcal{N}(\mathbf{x}_l; \boldsymbol{U}_{l-1}\mathbf{x}_{l-1}, \boldsymbol{\Sigma}_{l|l-1})\,.$$

---

[1]We omit normalizing constants. Equivalence and approximation are up to normalization in the following.

With the Cholesky decomposition (or eigen-decomposition) $\boldsymbol{\Sigma}_{l|l-1} = \boldsymbol{B}_l \boldsymbol{B}_l^T$, we design the prior conditioning layer $\mathrm{PC}_l$ as below, which is invertible between $\mathbf{x}_l$ and $(\mathbf{x}_{l-1}, \mathbf{z}_l)$:

$$\mathbf{x}_l = \mathrm{PC}_l(\mathbf{x}_{l-1}, \mathbf{z}_l) := \boldsymbol{U}_{l-1}\mathbf{x}_{l-1} + \boldsymbol{B}_l \mathbf{z}_l\,, \quad \mathbf{z}_l \sim \mathcal{N}(\mathbf{0}, \boldsymbol{I}_{d_l - d_{l-1}})\,. \tag{6}$$

We refer readers to Appendix B for proof of Lemma 2.1 and the invertibility in Equation 6.

When the prior is non-Gaussian or the pooling operators are nonlinear, there exists a *nonlinear* invertible prior conditioning operator $\mathbf{x}_l = \mathrm{PC}_l(\mathbf{x}_{l-1}, \mathbf{z}_l)$ such that $\mathbf{x}_l$ follows the prior conditional distribution $\rho(\mathbf{x}_l|\mathbf{x}_{l-1})$ given $\mathbf{x}_{l-1}$ and $\mathbf{z}_l \sim \mathcal{N}(\mathbf{0}, \boldsymbol{I}_{d_l - d_{l-1}})$. We can pre-train an invertible network to approximate this sampling process, and fix it as the prior conditioning layer.

**Invertible flow.** The invertible flow $F_l$ at level $l$ modifies the surrogate $\tilde{q}_l$ towards the target $q_l$. The more accurate the multiscale structure in Equation 3 is, the better $\tilde{q}_l$ approximates $q_l$, and the closer $F_l$ is to the identity map. Therefore, we parameterize $F_l$ by some flow-based generative model and initialize it as an identity map. In practice, we utilize the invertible block of Glow (Kingma & Dhariwal, 2018), which consists of actnorm, invertible $1 \times 1$ convolution, and affine coupling layer, and stack several blocks as the inverse flow $F_l$ in MsIGN.

**Overall model.** MsIGN is a bijective map between random noise inputs at different scales $\{\mathbf{z}_l\}_{l=1}^L$ and the finest-scale sample $\mathbf{x}_L$. The forward direction of MsIGN maps $\{\mathbf{z}_l\}_{l=1}^L$ to $\mathbf{x}_L$ as below:

$$\begin{aligned} \mathbf{x}_1 &= F_1(\mathbf{z}_1)\,, \\ \tilde{\mathbf{x}}_l = \mathrm{PC}_l(\mathbf{x}_{l-1}, \mathbf{z}_l)\,, \quad \mathbf{x}_l &= F_l(\tilde{\mathbf{x}}_l)\,, \quad 2 \leq l \leq L\,. \end{aligned} \tag{7}$$

As a flow-based generative model, sample generation as in Equation 7 and density evaluation $p_\theta(\mathbf{x})$ by the change-of-variable rule is accessible and fast for MsIGN. In scenarios when certain bound needs enforcing to the output, we can append element-wise output activations at the end of MsIGN. For example, image synthesis can use the sigmoid function so that pixel values lie in $[0, 1]$. Such activations should be bijective to keep the invertible relation between random noise to the sample.

## 2.3 MULTISCALE INVERTIBLE GENERATIVE NETWORK: TRAINING

Since the prior conditioning layer PC is pre-computed and the output activation $G$ is fixed, only the inverse flow $F$ contains trainable parameters in MsIGN. We train MsIGN with the following strategy so that the distribution $p_\theta$ of its output samples, where $\theta$ is the network parameter, can approximate the target distribution $q$ defined in Equation 1 well.

**Multi-stage training and interpret-ability.** The multiscale strategy in construction of MsIGN enables a coarse-to-fine multi-stage training. At stage $l$, we target at capturing $q_l$, and only train invertible flows before or at this level: $F_{l'}, l' \leq l$. Equation 4 implies that $q_l$ can be well approximated by the surogate $\tilde{q}_l$, which is the conditional upsampling from $q_{l-1}$ as in Equation 5. So we use $\tilde{q}_l$ to initialize our model by setting $F_{l'}, l' < l$ as the trained model at stage $l - 1$ and setting $F_l$ as the identity map. Our experiments demonstrate such multi-stage strategy significantly stabilizes training and improves final performance.

Figure 1 and Equation 7 imply that intermediate activations, i.e., $\tilde{\mathbf{x}}_l$ and $\mathbf{x}_l$, who are samples of predefined posterior distributions at the coarse scales (see Equation 5), are semantically meaningful and interpret-able. This is different from Glow (Kingma & Dhariwal, 2018), whose intermediate activations are not interpret-able due to the loss of spatial relation.

**Jeffreys divergence and importance sampling with the surrogate.** The KL divergence is easy to compute, and thus is widely used as the training objective. However, its landscape could admit local minima that don't favor the optimization. Nielsen & Nock (2009) suggests that $D_{\mathrm{KL}}(p_\theta \| q)$ is zero-forcing, meaning that it enforces $p_\theta$ be small whenever $q$ is small. As a consequence, mode missing can still be a local minimum, see Appendix C. Therefore, we turn to the Jeffreys divergence defined in Equation 2 which penalizes mode missing much and can remove such local minima.

Estimating the Jeffreys divergence requires computing an expectation with respect to the target $q$, which is normally prohibited. Since MsIGN constructs a good approximation $\tilde{q}$ to $q$, and $\tilde{q}$ can be constructed from coarser levels in multi-stage training, we do importance sampling with the

surrogate $\tilde{q}$ for the Jeffreys diveregence and its derivative (see Appendix D for detailed derivation):

$$D_{\mathrm{J}}(p_\theta \| q) = \mathbb{E}_{\mathbf{x} \sim p_\theta} \left[ \log \frac{p_\theta(\mathbf{x})}{q(\mathbf{x})} \right] + \mathbb{E}_{\mathbf{x} \sim \tilde{q}} \left[ \frac{q(\mathbf{x})}{\tilde{q}(\mathbf{x})} \log \frac{q(\mathbf{x})}{p_\theta(\mathbf{x})} \right] . \tag{8}$$

$$\frac{\partial}{\partial \theta} D_{\mathrm{J}}(p_\theta \| q) = \mathbb{E}_{\mathbf{x} \sim p_\theta} \left[ \left( 1 + \log \frac{p_\theta(\mathbf{x})}{q(\mathbf{x})} \right) \frac{\partial \log p_\theta(\mathbf{x})}{\partial \theta} \right] - \mathbb{E}_{\mathbf{x} \sim \tilde{q}} \left[ \frac{q(\mathbf{x})}{\tilde{q}(\mathbf{x})} \frac{\partial \log p_\theta(\mathbf{x})}{\partial \theta} \right] . \tag{9}$$

With the derivative estimate given above, we optimize the Jeffreys divergence by stochastic gradient descent. We remark that $\partial \log p_\theta(\mathbf{x}) / \partial \theta$ is computed by the backward propagation of MsIGN.

## 3 RELATED WORK

Invertible generative models (Deco & Brauer, 1995) are powerful exact likelihood models with efficient sampling and inference. They have achieved great success in natural image synthesis, see, e.g., Dinh et al. (2016); Kingma & Dhariwal (2018); Grathwohl et al. (2018); Ho et al. (2019); Chen et al. (2019b), and variational inference in providing a tight evidence lower bound (ELBO), see, e.g, Rezende & Mohamed (2015). In this paper, we propose a new multiscale invertible generative network (MsIGN) structure, which utilizes the invertible block in Glow (Kingma & Dhariwal, 2018) as building piece for the invertible flow at each scale. The Glow block can be replaced by any other invertible blocks, without any algorithmic changes. Different from Glow, different scales of MsIGN can be trained separately, and thus features in its intermediate layers can be interpreted as low-resolution approximation of the final high-resolution output. This novel multiscale structure enables better explain-ability of its hidden neurons and makes training much more stable.

Different from the image synthesis task where large amount of samples from target distribution are available, in Bayesian inference problems only an unnormalized density is available and i.i.d. samples from the posterior are the target. This paper's main goal is to train MsIGN to approximate certain high-$d$ Bayesian posteriors. Various kinds of parametric distributions have been proposed to approximate posteriors before, such as polynomials (El Moselhy & Marzouk, 2012), non-invertible generative networks (Feng et al., 2017; Hou et al., 2019), invertible networks (Rezende & Mohamed, 2015; Ardizzone et al., 2018; Kruse et al., 2019) and certain implicit maps (Chorin & Tu, 2009; Morzfeld et al., 2012). Generative modeling approach has the advantage that i.i.d. samples can be efficiently obtained by evaluating the model in the inference stage. However, due to the tricky non-convex optimization problem, this approach for both invertible (Chorin & Tu, 2009; Kruse et al., 2019) and non-invertible (Hou et al., 2019) generative models becomes increasingly challenging as the dimension grows. To overcome this difficulty, we propose (1) to use the Jeffreys divergence as loss function, which has fewer shallow local minima and better landscape compared with the commonly-used KL divergence (see Appendix C for a concrete example), and (2) to train MsIGN in a coarse-to-fine manner with coarse-scale solution serving as an initialization to fine-scale optimization problem. In Kruse et al. (2019), authors list some recent models for low-$d$ inverse problems. We remark that their formulation of posterior assumes no observation or model error in Equation 1, and is different from ours. See Appendix J for detailed discussion and experimental comparison.

Other than the generative modeling, various Markov Chain Monte Carlo (MCMC) methods have been the most popular in Bayesian inference, see, e.g., Beskos et al. (2008); Neal et al. (2011); Welling & Teh (2011); Chen et al. (2014; 2015); Cui et al. (2016). Particle-optimization-based sampling is a recently developed effective sampling technique with Stein variational gradient descent (SVGD) (Liu & Wang, 2016)) and many related works, e.g., Liu (2017); Liu & Zhu (2018); Chen et al. (2018; 2019a). The intrinsic difficulty of Bayesian inference displays itself as highly correlated samples, leading to undesired low sample efficiency, especially in high-$d$ cases. The multiscale structure and multi-stage strategy proposed in this paper can also benefit these particle-based methods, as we can observe that they benefit the amortized-SVGD (Feng et al., 2017; Hou et al., 2019) in Section 4.1.3. We leave a more thorough study of this topic as a future work.

Works in Parno et al. (2016); Matthies et al. (2016) utilize the multiscale structure in Bayesian inference and build generative models with polynomials. They suffer from exponential growth of parameter number for high-$d$ polynomial basis. The Markov property (Spantini et al., 2018) is used to alleviate this exponential growth. Different from these works, we leverage the great capacity of invertible generative networks to parametrize the high-$d$ distribution, and we design novel network architecture to make use of the multiscale structure. The multiscale structure is a more general

structure than commonly-used intrinsic low-$d$ structure (Spantini, 2017; Cui et al., 2016; Chen et al., 2019a), which assumes that the density of high-$d$ posterior concentrates in a low-$d$ subspace.

In the image synthesis task, this multiscale idea incorporates with various generative models. For example, Denton et al. (2015); Odena et al. (2017); Karras et al. (2017); Xu et al. (2018) uses it in generative adversarial networks (GANs) to grow a high-resolution image from low-resolution ones. But the lack of invertibility in these models makes it difficult for them to apply to Bayesian inference problems. Invertible generative models like Dinh et al. (2016); Kingma & Dhariwal (2018); Ardizzone et al. (2019) adopted this multiscale idea, but their multiscale strategy is not in the spatial sense: the intermediate neurons are not semantically interpret-able, as we show in Figure 6.

## 4 EXPERIMENT

We study two high-$d$ Bayesian inverse problems (BIPs) known to have at least two equally important modes in Section 4.1 as test beds for distribution approximation and multi-mode capture: one with true samples available in Section 4.1.1; one without true samples but close to real-world applications in subsurface flow in Section 4.1.2. We also report the ablation study of MsIGN in Section 4.1.3. In addition, we apply MsIGN to the image synthesis task to benchmark with flow-based generative models and demonstrate its interpret-ability in Section 4.2. We adopt the invertible block in Glow (Kingma & Dhariwal, 2018) as the building piece, and stack several of them to build our invertible flow $F$. We utilize average pooling with kernel size 2 and stride 2 as our pooling operator $\mathcal{A}$.

### 4.1 BAYESIAN INVERSE PROBLEMS

Sample $\mathbf{x}$ of our target posterior distribution $q$ is a vector on a 2-D uniform $64 \times 64$ lattice, which means the problem dimension $d$ is 4096. Every $\mathbf{x}$ is equivalent to a piece-wise constant function on the unit disk: $\mathbf{x}(s)$ for $s \in \Omega = [0, 1]^2$, and we don't distinguish between them thereafter. We place a centered Gaussian with a Laplacian-type covariance as the prior: $\mathcal{N}\left(0, \beta^2(-\mathbf{\Delta})^{-1-\alpha}\right)$, which is very popular in geophysics and electric tomography. See Appendix E for problem settings in detail.

The key to guarantee the multi-modality of our posteriors is the symmetry. Combining properties of the prior defined above and the likelihood defined afterwards, the posterior is mirror-symmetric: $q(\mathbf{x}(s_1, s_2)) = q(\mathbf{x}(s_1, 1 - s_2))$. We carefully select the prior and the likelihood so that our posterior $q$ has at least two modes. They are mirror-symmetric to each other and possess equal importance.

As in Figure 1, we plan to learn our 4096-D posteriors at the end of $L = 6$ levels, and set problem dimension at each level as $d_l = 2^l * 2^l = 4^l$. The training follows our multi-stage strategy, and the first stage $l = 1$ is initialized by minimizing the Jeffreys divergence without importance sampling, because samples to $q_1$ is available since $d_1 = 4$ is relatively small. See Appendix E for details.

We compare MsIGN with representatives of major approaches: amortized-SVGD (short as A-SVGD) (Feng et al., 2017) and Hamilton Monte Carlo (short as HMC) (Neal et al., 2011), for high-$d$ BIPs, see our discussion in Section 3. We measure the computational cost by the number of forward simulations (nFSs), because running the forward simulation $\mathcal{F}$ occupies most training time, especially in Section 4.1.2. We budget a same nFS for all methods for fair comparison.

#### 4.1.1 SYNTHETIC BAYESIAN INVERSE PROBLEMS

This problem allows access to ground-truth samples so the comparison is clear and solid. The forward process is given by $\mathcal{F}(\mathbf{x}) = \langle \boldsymbol{\varphi}, \mathbf{x} \rangle^2 = (\int_\Omega \varphi(s)\mathbf{x}(s)\mathrm{d}s)^2$, where $\varphi(s) = \sin(\pi s_1)\sin(2\pi s_2)$. Together with the prior, our posterior can be factorized into one-dimensional sub-distributions, namely $q(\mathbf{x}) = \prod_{k=1}^d q_k(\langle \boldsymbol{w}_k, \mathbf{x} \rangle)$ for some orthonormal basis $\{\boldsymbol{w}_k\}_{k=1}^d$. This property gives us access to true samples via inversion cumulative function sampling along each direction $\boldsymbol{w}_k$. Furthermore, these 1-D sub-distributions are all single modal except that there's one, denoted as $q_{k^*}$, with two symmetric modes. In other words, the marginal distribution along $\boldsymbol{w}_{k^*}$ is double-model and the rest are uni-model. This confirms our construction of two equally important modes. See Appendix E for more details in problem settings. The computation budget is fixed at $8 \times 10^5$ nFSs.

**Multi-mode capture.** To visualize mode capture, we plot the marginal distribution of generated samples along the critical direction $\boldsymbol{w}_{k^*}$, which by construction is the source of double-modality of the posterior. The (visually) worst one in three independent experiments is shown in Figure 2(a).

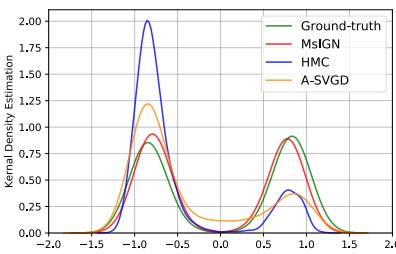 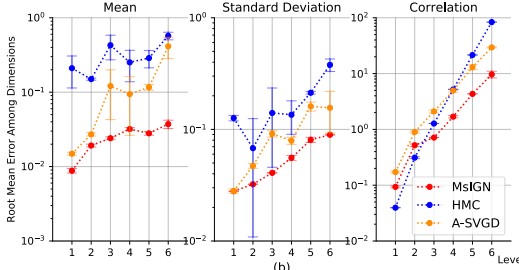

Figure 2: Results of the synthetic BIP. (a): Distribution of 2500 samples along the critical direction $\boldsymbol{w}_{k^*}$. MsIGN is more robust in capturing both modes, its samples are more balanced. (b): Error mean and its 95% confidence interval. MsIGN is more accurate in distribution approximation, especially at finer scale when the problem dimension is high. The margin is statistical significant as shown by the confidence interval. For more experimental results, please refer to Appendix F.

**Distribution approximation.** To measure distribution approximation, we report the error of mean, variance and correlation at or between all sub-distributions, as well as the Jeffreys divergence. Thanks to the factorization property, we compare the mean, variance and correlation estimate with theoretical ground-truths, and report the root mean square of error at all

| Method | MsIGN | A-SVGD (Feng et al., 2017) |
|---|---|---|
| Error | 56.77±0.15 | 3372±21 |

Table 1: Distribution approximation error by Jeffreys divergence with the target posterior in three independent runs

dimensions in Figure 2(b). For MsIGN and A-SVGD that gives access to not only samples but also density, we also report the Monte Carlo estimates of the Jeffreys divergence with the target posterior in Table 1. We can see that MsIGN has superior accuracy in approximating the target distribution.

### 4.1.2 ELLIPTIC BAYESIAN INVERSE PROBLEMS

This problem originates from geophysics and fluid dynamics. The forward model is given by linear measurement of the solution to an elliptic partial differential equation associated with $\mathbf{x}$. We define

$$\mathcal{F}(\mathbf{x}) = \left[ \int_\Omega \varphi_1(s)\mathbf{u}(s)\mathrm{d}s \quad \int_\Omega \varphi_2(s)\mathbf{u}(s)\mathrm{d}s \quad \dots \quad \int_\Omega \varphi_m(s)\mathbf{u}(s)\mathrm{d}s \right]^T ,$$

where $\varphi_k$ are fixed measurement functions, and $\mathbf{u}(s)$ is the solution of

$$-\nabla \cdot \left( e^{\mathbf{x}(s)} \nabla \mathbf{u}(s) \right) = f(s) \, , s \in \Omega \, , \quad \text{with boundary condition } \mathbf{u}(s) = 0 \, , s \in \partial\Omega \, . \quad (10)$$

This model appears frequently in real applications. For example, x, u can be seen as permeability field and pressure in geophysics. However, there is no known access to true samples of $q$. Again the trick of symmetry introduced in Section 4.1 and explained in Appendix E guarantees at least two equally important modes in the posterior. We put a $5 \times 10^5$-nFS budget on our computation cost.

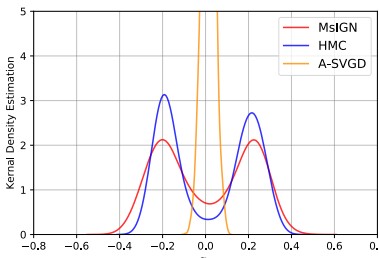 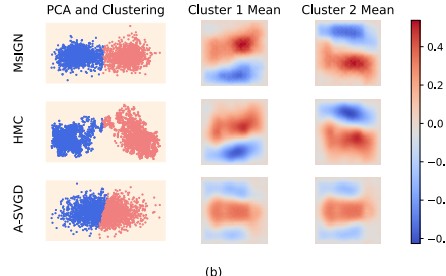

Figure 3: Results of the elliptic BIP. (a): Distribution of 2500 samples along a critical direction. MsIGN and HMC capture two modes in this marginal distribution, but A-SVGD fails. (b): Clustering result of 2500 samples. Samples of MsIGN are more balanced between two modes. The similarity of the cluster means of MsIGN and HMC implies that they both are likely to capture the correct modes. For more experimental results, please refer to Appendix I.

**Multi-mode capture.** Due to lack of true samples, we check the marginal distribution of the posterior along eigen-vectors of the prior, and pick a particular one to demonstrate that we can capture double modes in Figure 3(a). We also confirm the capture of multiple modes by embedding samples

by Principle Component Analysis (PCA) to a 2-D space. We report the clustering (by K-means) result and means of each cluster in Figure 3(b), where we can see that A-SVGD failed to capture the two symmetric modes, while MsIGN has a more balanced capture of the symmetric posterior.

### 4.1.3 ABLATION STUDY OF ARCHITECTURE DESIGN AND TRAINING STRATEGY

We run extensive experiments to study the effectiveness of the network architecture and training stragtegy of MsIGN, see Figure 4. We refer to Appendix G for details in setting and more results.

**Network architecture.** We replace the prior conditioning layer by two direct alternatives: a stochastic nearest-neighbor upsampling layer (model denoted as "MsIGN-SNN"), or the split and squeeze layer in Glow design (now the model is essentially Glow, so we also denote it as "Glow").

Figure 4(a) shows that the prior conditioning layer design is crucial to the performance of MsIGN on both problems, because neither "MsIGN-SNN" nor "Glow" has a successful mode capture.

**Training strategy.** We study the effectiveness of the Jeffreys divergence objective and multi-stage training. We try substituting the Jeffreys divergence objective (no extra marks) with the KL divergence (model denoted with a string "-KL") or kernelized Stein discrepancy (which resumes A-SVGD algorithm, model denoted with a string "-AS"), and switching between multi-stage (no extra marks) or single-stage training (model denoted with a string "-S"). We remark that single-stage training using Jeffreys divergence is infeasible because of the difficulty to estimate $D_{\mathrm{KL}}(q\|p_\theta)$.

Figure 4(b) and (c) show that, all models trained in the single-stage manner ("MsIGN-KL-S", "MsIGN-AS-S") will face mode collapse. We also observe that our multi-stage training strategy can benefit training with other objectives, see "MsIGN-KL" and "MsIGN-AS".

We also notice that the Jeffreys divergence leads to a more balanced samples for these symmetric problems, especially for the complicated elliptic BIP in Section 4.1.2.

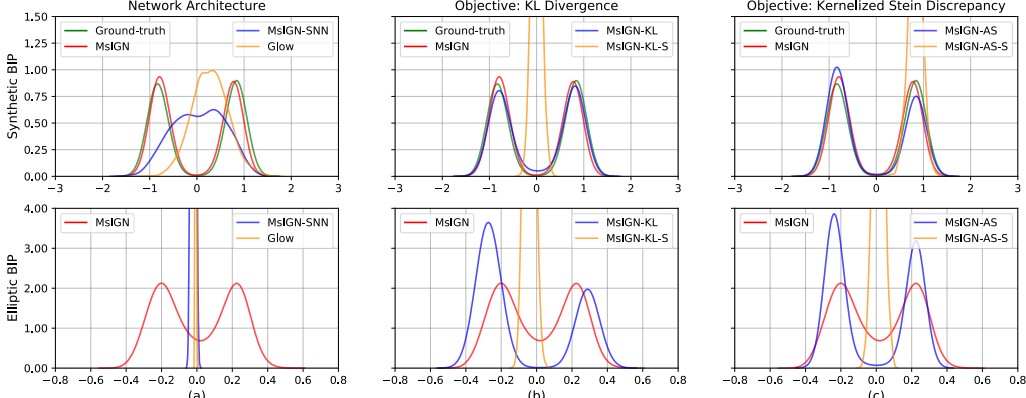

Figure 4: Ablation study of the network architecture and training strategy. "MsIGN" means our default setting: training MsIGN network with Jeffreys divergence and multi-stage strategy. Other models are named by a base model (MsIGN or Glow), followed by strings indicating its variance from the default setting. For example, "MsIGN-KL" refers to training MsIGN network with single KL divergence in a multi-stage way, while "MsIGN-KL-S" means traininng in a single-stage way.

### 4.2 IMAGE SYNTHESIS TASK

We train our MsIGN architecture with maximum likelihood estimation to benchmark with other flow-based generative models. The prior conditional distribution $\rho(\mathbf{x}|\mathbf{x}_c)$ is modeled by a simple Gaussian with a scalar matrix as its covariance and is learned from a training set. We refer readers to Appendix H for more experimental details, and to Appendix I for additional results.

We report the bits-per-dimension value with our baseline models of flow-based generative networks in Table 2. Our MsIGN is superior in number and also is more efficient in parameter size: for example, MsIGN uses $24.4\%$ fewer parameters than Glow for CelebA 64, and uses $37.4\%$ fewer parameters than Residual Flow for ImageNet 64.

In Figure 5, we show synthesized images of MsIGN from CelebA 64 dataset, and linear interpolation of real images in the latent feature space. In Figure 6, we visualize internal activations at checkpoints of the invertible flow at different scales which demonstrates the interpret-ability of MsIGN.

Table 2: Bits-per-dimension value comparison with baseline models of flow-based generative networks. All models in this table do not use "variational dequantization" in Ho et al. (2019). *: Score obtained by our own reproducing experiment.

| Model | MNIST | CIFAR-10 | CelebA 64 | ImageNet 32 | ImageNet 64 |
|---|---|---|---|---|---|
| Real NVP (Dinh et al., 2016) | 1.06 | 3.49 | 3.02 | 4.28 | 3.98 |
| Glow (Kingma & Dhariwal, 2018) | 1.05 | 3.35 | 2.20* | 4.09 | 3.81 |
| FFJORD (Grathwohl et al., 2018) | 0.99 | 3.40 | – | – | – |
| Flow++ (Ho et al., 2019) | – | 3.29 | – | – | – |
| i-ResNet (Behrmann et al., 2019) | 1.05 | 3.45 | – | – | – |
| Residual Flow (Chen et al., 2019b) | 0.97 | **3.28** | – | **4.01** | 3.76 |
| **MsIGN** (Ours) | **0.93** | **3.28** | **2.15** | 4.03 | **3.73** |

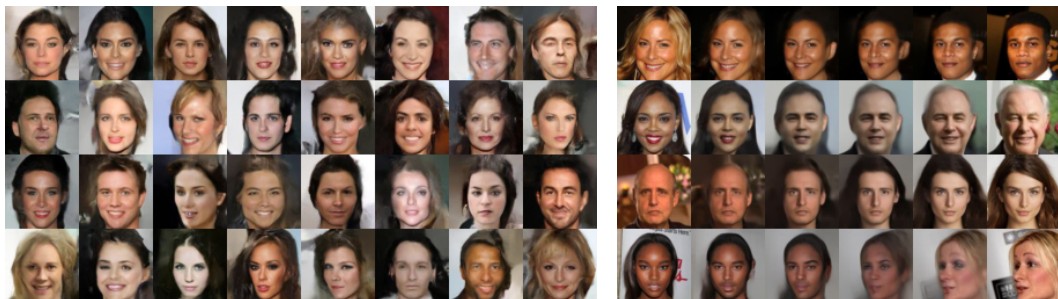

Figure 5: Left: Synthesized CelebA 64 images with temperature 0.9. Right: Linear interpolation in latent space shows MsIGN's parameterization of natural image manifold is semantically meaningful.

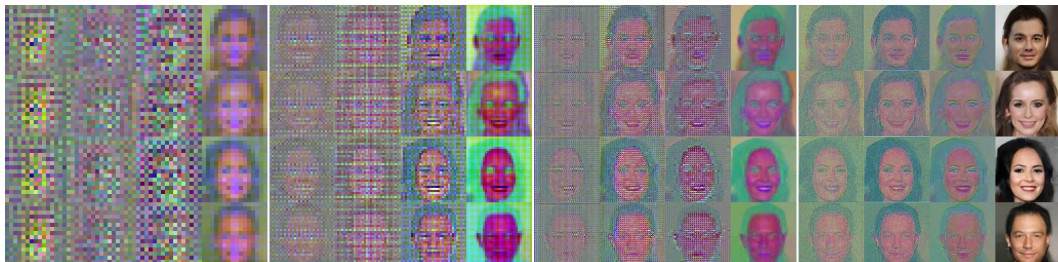

Figure 6: Visualization of internal activation shows the interpret-ability of MsIGN hidden neurons. From left to right, we show how MsIGN progressively generates new samples in high resolution by taking snapshots at internal checkpoints. See Appendix I for details.

## 5 CONCLUSION

For high-dimensional Bayesian inference problems with multiscale structure, we propose Multiscale Invertible Generative Networks (MsIGN) and associated training algorithms to approximate the high-dimensional posterior. In this paper, we demonstrate the capability of this approach in high-dimensional (up to 4096 dimensions) Bayesian inference problems with spatial multiscale structure, leaving several important directions as future work. The network architecture also achieves the state-of-the-art performance in various image synthesis tasks. We plan to apply this methodology to other Bayesian inference problems, for example, Bayesian deep learning with multiscale structure in model width or depth (e.g., Chang et al. (2017); Haber et al. (2018)) and data assimilation problem with multiscale structure in the temporal variation (e.g., Giles (2008)). We also plan to develop some theoretical guarantee of the posterior approximation performance for MsIGN.

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
