# OpenReview forum: "Multiscale Invertible Generative Networks for High-Dimensional Bayesian Inference"
_ICLR.cc/2021/Conference — Reject_

### Official Review · AnonReviewer3 · 2020-10-27
**I believe the paper has its merits, but frankly I could not understand much**

**Rating:** 5
**Confidence:** 1

**Review:**

While the use of an hierarchical flow-based model to obtain samples of finer scales seems interesting, I have to admit that I did not understand much of the setup the authors propose. The methodology section is too dense and many of their claims are not really motivated at all.

I do not see this paper accepted the way it is, but it might be true that a person more expert in flow-based models and inference than me thinks completely the opposite.

---

> ### Author Response · Authors · 2020-11-17
> **Comment to Reviewer 3**
>
> Thank you for your honest comment.
>
> Our work targets at the Bayesian inference problems, where we want to generate samples from a distribution, given its unnormalized density. Such problem is very challenging when the sample dimension is high and when there are multiple modes in the distribution. Our MsIGN proposes a way to detect multiple modes and approximate the target distribution when the problem demonstrates certain multiscale structure, even if the dimension is rather high.
>
> Now we've updated our manuscript to enhance its clarity. Meanwhile, we will be very glad to discuss your questions and concerns.

---

### Official Review · AnonReviewer2 · 2020-10-28
**The lack of clarity is in the way of making this a good paper that I am willing to accept. (Update: Marginal accept)**

**Rating:** 6
**Confidence:** 3

**Review:**

# Paper Summary
This paper presents a model and a corresponding training approach for multi-scale invertible models. The presented model is defined on multiple scales with information on finer scales being conditioned on coarser scales. Data generation is hence done sequentially from a coarser to finer scale. The authors argue that this multi-scale sampling helps in addressing the curse of dimensionality problem by allowing to sample from high density regions more efficiently.

As a core part of the model, the authors present a linear layer that combines information from a coarse scale and a noise vector to predict the data on a finer scale. This prior conditioning layer, as the authors call it, exploits the fact that the downsampling operation from a finer scale to a coarser scale is a linear operation. Under the assumption that the fine scale data follows a multivariate normal distribution, the downsampling operation can be analytically inverted.

Training is done by minimizing Jeffreys divergence, i.e. the loss is calculated bi-directinally in the forward and the backward direction of the model, akin to Ardizzone et al. (2019). The authors argue that Jeffreys divergence is less prone to the mode dropping that can happen when training with KL divergence. The authors justify this argument by citing Nielsen & Nock (2009), by doing ablation studies on toy data, and by giving a toy example in Appendix C.

Due to the multi-scale structure and the fact that the coarse scale data can be generated from a fixed corruption process (the downsampling model), it is possible to train this model by decoupling the different scales. The authors suggest to train the model from a coarser to finer scale in a multi-stage process. The authors claim the superiority of this approach to regular end-to-end training.

In an extensive experimental section, the authors evaluate their approach on two 'Bayesian Inverse Problems', as well as on generative modeling of the Celeb A 64 data set. In further experiments, the authors perform ablation studies on a synthetic data set in order to justify their presented approach. In all experiments, the presented approach yields the highest performance.

# Assessment Summary

The presented model and training approach exploit the hierarchical nature of data in a smart way. This in turn results in good performance on a number of data sets. The experimental section is rigorous and convincing. These aspects of the work should put me in favor of acceptance if it weren't for the lack of clarity in places (see below). In its current state the paper does not explain the method well enough in order to get a full understanding of the approach. I therefore tend to reject the paper at this stage. With appropriate improvements in the writing I would be happy to improve my score.

# Positives
- A multi-scale approach is a reasonable choice in many real world applications. The presented model exploits our world knowledge by utilizing a fixed downsampling operator to generate views of the data on different scales. This kind of inductive bias is simple to interpret and it appears to be a promising aid in model training.
- The experimental section is extensive. The authors evaulate their approach on multiple problems, synthetic and real world. Especially the ablation study should be pointed out.
- The ablation study aims to confirm the authors claims that the presented approach results in better mode capture. To do that the authors compare different model architectures, different training losses, and the multi-stage training to end-to-end training. Their results suggest that the multi-scale model trained in multiple stages using Jeffreys divergence indeed performs best.

# Negatives
- One of the biggest issues in this paper is the lack of clarity in places, which in turn makes the work hard to digest. This is often owed either to a poor choice of notation or to abbreviated explanations. Especially the methods section should be pointed out in this regard. I will give a few examples below.
	- In equation (1) the authors surrogate $q(x|y)$ with $q(x)$. This choice makes it difficult to follow equations (5) and (6) as it is not clear anymore where we are dealing with marginal or conditional probabilities.
	- The use of $\rho(x)$ and $L(y|x)$ does not follow the ICLR style guide for probabilities.
	- Section 2.1 is not clear. Here, the choice of abbreviations in (5) and (6) hinders an understanding. It is not made clear which role variable y plays in the network. It should be made more clear by the authors that the forward process of the multiscale model is sampling from the posterior distribution.
	- Figure 1 only shows the sampling process, but does not help to understand the training. $\tilde{x}_l$ is represented with smaller images than $x_l$ although both have the same dimensionality.
	- Section 2.3 discusses biefly the use of Jeffreys divergence with importance sampling for training the model. The authors do not make clear what $p(x)$ and $q(x)$ stand for in this context. As such it is not possible to make out how to train the model with the suggested loss.
- The prior conditioning layer (upsampling layer) exploits the fact that the downsampling operation is linear and the prior over fine-scale data is assumed to be a multivariate normal distribution. As such, the upsampling can be implemented with a linear function of the coarse data and a noise vector. It is well known that this kind of inversion, although analytically correct, is computationally infeasible in high dimensional data. The authors do not address this issue in their text.
- With respect to the multi-scale structure of the model, the paper does not position itself well in the literature. Invertible models such as in Dinh et al. (2016) already use a multi-scale structure. The authors should point out the difference of their approach more explicitly. Using explicit downsampling as an inductive bias was used successfully before for training GANs in Denton et al. (2015). This work might be a good starting point to discover other related multi-scale approaches.


# Recommendations
- Figure 1 could be improved to visualise the overall approach better. It would be worth visualising the sampling process for estimating Jeffreys divergence. Further, I would suggest to use an example from the Celeb A data set in order to get a better understanding of the intermediate scales. In addition, the downsampling operation $\mathcal{A}$ could be emphasided in this figure.
- Notation should be improved and made more consistent. Is there a way to reduce the amount of variables used? Is it possible to use consistent notation to denote probabilities?
- Section 2.1 should be clarified. Is there a simpler way to explain this? For example by exploiting the Markov property of the model.

# Questions
- In Lemma 2.1 it is assumed that the fine-scale signal follows a multivariate normal distribution. Is this a reasonable assumption? Does this not contradict the idea of multiple modes in the data?
- Figure 6: Why do the interemediate coarser scale samples not look like downsampled versions of the high resolution image?
- What happens if the models on each scale are trained completely decoupled? Would it be fair to call the model a stacked super-resolution model?

# References

Ardizzone, L., Kruse, J., Wirkert, S., Rahner, D., Pellegrini, E. W., Klessen, R. S., & Köthe, U. (2018). Analyzing inverse problems with invertible neural networks.

Denton, E. L., Chintala, S., & Fergus, R. (2015). Deep generative image models using a￼ laplacian pyramid of adversarial networks.

Dinh, L., Sohl-Dickstein, J., & Bengio, S. (2016). Density estimation using real nvp.

Nielsen, F., & Nock, R. (2009). Sided and symmetrized Bregman centroids.

---------------------------------------------------------

# Review Update
Based on the response and update from the authors I will improve my score from '5: Marginally below acceptance threshold' to '6: Marginally above acceptance threshold'. I generally believe that the presented approach merits publication. The methodology is sound and the experiments are convincing. I appreciate the fact that the authors made an effort to improve the writing.

---

> ### Author Response · Authors · 2020-11-17
> **Comment to Reviewer 2**
>
> Thank you for your detailed review, comments and suggestions.
>
> We've updated our manuscript, especially for the methodology part, to increase the clarity of this work.
>
> #### Regarding the negatives:
>
> - $q(x)$ vs $q(x|y)$: The measurement data $y$ only takes the part of defining the target distribution $q(x|y)$ in MsIGN, so we decided to drop it for notation simplicity. In fact, in other literatures in Bayesian inference problems like Moselhy et al., (2012) and Chen et al., (2019), authors used a similar abbreviation to the target posterior distribution. We updated our wording to make this notation clearer to readers in the beginning of Section 2.
>
> - Prior conditioning operator incomputable: The prior conditioning layer only needs to be computed once and will be fixed thereafter. Thus although sometimes the dimension gets too high and the inversion might be very expensive, it might still be acceptable given that we only needs to compute it once. Furthermore, if the dimension gets so high such that one-time evaluation is infeasible, we can consider performing low-rank approximation of the analytical solution. Though the prior conditioning operator is not exact, the following unit of invertible flow will correct this approximation by training.
>
> - Related work for the multi-scale design: Since we mainly focus on the task of Bayesian inference, in Section 3 we mainly listed related work for this task. We thank the reviewer for reminding us of the related works in image synthesis task, and have revised this section in the manuscript. The RealNVP model in Dinh et al., (2016) and Glow model in Kingma & Dhariwal., (2018) do used a multi-scale design, but the main difference is that 1) their multi-scale design doesn’t have spatial semantic meaning. The internal neuron gets a reduced of dimension, but this reduction has no spatial property. 2) Due to 1), RealNVP and Glow can’t be trained in a multi-stage fashion, and generate samples in a coarse-to-fine manner. The GAN model in Denton et al., (2015) is a generative adversial network that is not invertible, which is hard to do density evaluation. And therefore, the lack of invertibility in this models makes it difficult to be applied to the Bayesian inference problems. We appended more related work discussion in Section 3.
>
> #### Regarding the questions
>
> - Assumption of the multi-variate normal distribution: In the image synthesis task, since we lack of the information of how the distribution of images looks like a priori, we assume the prior of the images to be a multi-variate normal distribution. This doesn’t contradict with the multi-modal nature of the data because it only enforces a prior (and in practice a very weak one) for the data. The prior helps us in building a prior conditioning distribution to guide us in refining a low-resolution image to a high-resolution one, and it won’t hurt the multi-modality.
>
> - Figure 6: We use the model trained on low-resolution data to initialize the high-resolution model. Parameters from the low-resolution model will not be frozen in the following training, and therefore, the output at low-resolution stage might be different from that of the original low-resolution model. As shown in Figure 6, they are most likely some linear mix of the original output along the color channel. We can still see that the spatial feature of the intermediate coarser scale samples resembles that of the finer scale samples.
>
> - Decoupled training and a stacked super-resolution model: For the image synthesis task, it is fair to call our model a stacked super-resolution model. In fact, we also report that given the data set of natural images, each scale of our model can be trained separately. However, we focus on solving Bayesian inverse problems using MsIGN, where true samples are not available in training. In this case, we can’t train each scale of MsIGN separately. For high-$d$ Bayesian inverse problem with multiple modes, it is crucial to have the coarse-scale model in hand to optimize the Jeffreys divergence by importance sampling, and initialize the fine-scale model. In this sense, it may be appropriate to call our MsIGN a stacked super-resolution model.
>
> #### References
>
> El Moselhy, Tarek A., and Youssef M. Marzouk. "Bayesian inference with optimal maps." Journal of Computational Physics 231.23 (2012): 7815-7850.
>
> Chen, Peng, et al. "Projected Stein variational Newton: A fast and scalable Bayesian inference method in high dimensions." Advances in Neural Information Processing Systems. 2019.
>
> Kingma, Durk P., and Prafulla Dhariwal. "Glow: Generative flow with invertible 1x1 convolutions." Advances in neural information processing systems. 2018.

---

### Official Review · AnonReviewer1 · 2020-10-30
**Multiscale Invertible Generative Networks for High-Dimensional Bayesian Inference**

**Rating:** 6
**Confidence:** 3

**Review:**

The authors propose a multi-scale method to design invertible network and apply it to bayesian inverse problems.

The method itself is quite concise and beautiful in its simplicity (a nice case of “obvious once you see it”), and looks rather simple to implement. The results are also rather promising.

With that said, there are some steps that are lacking. Notably the results leave some things open (e.g. the SotA-ness is highly debatable given more recent methods such as routing transformers) and for the Bayesian inverse problems it’s hard to judge given that the only baselines are “in house” rather than externally defined. I also find the theoretical exposition to be a bit hard to follow, especially given the generous use of appendices.

I have some further comments:

* The existence of a prior conditioning operator in the general case seems somewhat debatable (consider odd cases like singular distributions, distributions with weird topologies and discrete distributions), I’m quite sure you need at least some further conditions. I’d also be fine with a “if it exists” style statement.
* The optimisation described in appendix D is quite critical to the method as is (especially since using Jeffreys divergence is even mentioned in the abstract as a main point) and should be in the body of the article.
* The statement “the subsurface flow simulation at the scale of meters approximates well the simulation at the scale of centimeters” is more or less repeated twice verbatim. It’s also not the most accessible example.
* Spelling errors like dimebsion, groundtruth. Overall the language could use some polishing throughout the article.
* All figures should be in vector format
* The statement “ q(x(s1, s2)) = q(x(s1, 1 − s2))” does not afaik imply more than one mode, consider e.g. a mode centered at s2=0.5.
* Showing the “worst of 3” seems rather dishonest. Why was this done?
* Table 2 is quite lacking, several SotA methods are not in there (which makes this method not SotA).

---

> ### Author Response · Authors · 2020-11-17
> **Comment to Reviewer 1**
>
> Thanks for your careful review and detailed comments. We've updated our manuscript.
>
> Here are our reply to your comments:
>
> - The existence of a prior conditioning operator: When the distribution is odd, for example a singular distribution, the prior conditioning operator will exist but not be necessarily bijective or invertible. However, in Bayesian inference, the posterior distribution has a well-defined density in general. Besides, in those odd cases, we can still construct a prior conditioning operator that approximates the distribution well, rather than exactly captures it. This approximation will be refined by the following invertible flow unit. Theoretically not every distribution has a well-defined prior conditioning operator, but in the practice of Bayesian inference problems with multiscale structure, we expect this issue won’t be a trouble.
>
> - Optimization in Appendix D, repetition, spelling error, figures in vector format: We thank you very much for the suggestions. We’ve revised our manuscript accordingly.
>
> - “$q(x(s1, s2))=q(x(s1, 1-s2))$” does not imply multiple modes: Yes, “$q(x(s1, s2))=q(x(s1, 1-s2))$” is not a sufficient condition to the existence of multiple modes. This design only ensures that we will have a symmetric posterior distribution. To construct a posterior with multiple modes, we need to carefully choose the hyper-parameter $\alpha$, $\beta$, $\sigma_\varepsilon$, as well as the measurement functions $\varphi$, and the force term $f$. To certify the multi-modality, we run multiple gradient ascent searching of the posterior, starting from different initial points. We observe that they all converge to two mutually symmetric points $\mathbf{x}^*$ and $\tilde{\mathbf{x}}^*$. We also visualize the 1D landscape profile of the posterior q on the line passing through $\mathbf{x}^*$ and $\tilde{\mathbf{x}}^*$. The profile also shows a clear double-modal feature. We appended more necessary details in our manuscript, see Section 4.1 and Appendix E.
>
> - Showing the worst of 3: For Figure 2(a), we repeat the experiment for 3 times for robustness purpose. Different runs (of a same method) may capture different mode, and thus can be difficult or confusing to visualize. Therefore based on this consideration we only plot 1 out of these 3 runs, and our choice is the the worst of these 3.
>
> - SOTA claim and Table 2: Yes, it may be inadequate to claim the SOTA among all solutions to the image synthesis task. However, to our knowledge, MsIGN gives the best bit-per-dimension value among flow-based generative models. Besides, Routing transformer is an auto-regressive model and has much more number of parameters than ours (80M vs 33M for the ImageNet-64 data set). We also remark that the main focus of MsIGN is to solve the Bayesian inference problems here. We revised our wording to "achieves superior performance in bits-per-dimension among our baseline models, like Glow (Kingma & Prafulla., 2018), FFJORD (Grathwohl et al., 2018), Flow++ (Ho et al., 2019), i-ResNet(Behrmann et al., 2019), and Residual Flow (Chen et al., 2019)" in the updated version.
>
> #### References
>
> Kingma, Durk P., and Prafulla Dhariwal. "Glow: Generative flow with invertible 1x1 convolutions." Advances in neural information processing systems. 2018.
>
> Grathwohl, Will, et al. "Ffjord: Free-form continuous dynamics for scalable reversible generative models." arXiv preprint arXiv:1810.01367 (2018).
>
> Ho, Jonathan, et al. "Flow++: Improving flow-based generative models with variational dequantization and architecture design." arXiv preprint arXiv:1902.00275 (2019).
>
> Behrmann, Jens, et al. "Invertible residual networks." International Conference on Machine Learning. 2019.
>
> Chen, Ricky TQ, et al. "Residual flows for invertible generative modeling." Advances in Neural Information Processing Systems. 2019.

---

### Author Response · Authors · 2020-11-24
**Response and Revision**

We thank reviewers for their insightful feedback, appreciation and suggestions to our work! The manuscript (both the main body and the appendices) has been revised with the following major changes to incorporate the comments.

(Here Rx means Reviewer x for x=1,2,3)


- Based on R1 and R2’s comments and suggestions about the clarity issue, we have revised our Section 1 and 2. Notations are made clearer at the beginning of both sections, and equations are followed by more detailed explanation, especially in Section 2.1. We also revised the font style to follow the ICLR guideline as mentioned by R2. Images are re-generated to a vector format, if possible.


- Based on R1’s comments on the SotA claim, we revised our wording and do not claim SotA now.


- Based on R1’ question on the multi-modality design of the test bed, we revised our wording in Section 4.1 to make clear that “q(x(s1, s2)) = q(x(s1, 1-s2))” is not the sufficient condition to make the test bed multi-modal, but a helpful condition to generate a multi-modal testbed, and to make sure modes are mirror-symmetric, and hence balanced to each other. We also revised our Appendix E.1 to elaborate on this issue.


- Based on R2’s suggestion, we revised our Figure 1. We still decided to use samples from the Bayesian inverse problem as an example, because we wanted to emphasize our main target task of Bayesian inference.


- Based on R2’s comment, we also moved part of Appendix D to Section 2.3 to make the description of our training strategy more complete.


- Based on R2’s suggestion on more comprehensive literature review, we revised and appended new content in Section 3.


- Based on R2’ suggestion on the highlight of the ablation study, we appended more description on our ablation study at Section 4.1.3, in order to give a more detailed discussion in this section. We also revised our appendix G in better support of this section.


The appendix has been greatly revised to give clearer explanation and support to our paper.

---

### Decision · Program_Chairs · 2021-01-07
**Final Decision**

**Decision:**

Reject

**Comment:**

The paper considers sample generation in high-dimensional bayesian inference and proposes a multi scale procedure that performs coarse-to-fine multi-stage training and enables interpretability of intermediate activations at coarse scales.  The method is simple, elegant and addresses a very important bottleneck of high-dimensional bayesian inference. The clarity of the paper has been greatly improved based on the reviewers suggestions.

However some concerns remain regarding the evaluation that are needed to clearly demonstrate the value of the approach. In particular, it would be important to assess the impact of the number of levels, and how quickly the dimensions grow from one level to the next. The number of forward simulations does not provide a sufficient picture of the computational cost of the approaches. It would be also important to provide wall-clock time.  Figure 2a should also provide the best of 3  independent experiments,  or better, more experiments should be run, and curves with shaded areas should be provided so one could visualize variability w.r.t runs.